# The Long-Jumping of African Swine Fever: First Genotype II Notified in Sardinia, Italy

**DOI:** 10.3390/v16010032

**Published:** 2023-12-23

**Authors:** Silvia Dei Giudici, Federica Loi, Sonia Ghisu, Pier Paolo Angioi, Susanna Zinellu, Mariangela Stefania Fiori, Francesca Carusillo, Diego Brundu, Giulia Franzoni, Giovanni Maria Zidda, Paolo Tolu, Ennio Bandino, Stefano Cappai, Annalisa Oggiano

**Affiliations:** 1Laboratory of Virology, Deapartment of Animal Health, Istituto Zooprofilattico Sperimentale della Sardegna, 07100 Sassari, Italy; silvia.deigiudici@izs-sardegna.it (S.D.G.); pierpaolo.angioi@izs-sardegna.it (P.P.A.); susanna.zinellu@izs-sardegna.it (S.Z.); mariangela.fiori@izs-sardegna.it (M.S.F.); giulia.franzoni@izs-sardegna.it (G.F.); annalisa.oggiano@izs-sardegna.it (A.O.); 2Osservatorio Epidemiologico Veterinario Regionale della Sardegna, Istituto Zooprofilattico Sperimentale della Sardegna, 09125 Cagliari, Italy; stefano.cappai@izs-sardegna.it; 3Diagnostic Laboratories, Istituto Zooprofilattico Sperimentale della Sardegna, 08100 Nuoro, Italy; sonia.ghisu@izs-sardegna.it (S.G.); francesca.carusillo@izs-sardegna.it (F.C.); diego.brundu@izs-sardegna.it (D.B.); ennio.bandino@izs-sardegna.it (E.B.); 4Azienda Sanitaria Locale della Sardegna, 08100 Nuoro, Italy; giovannimaria.zidda@aslnuoro.it (G.M.Z.); paolo.tolu@aslnuoro.it (P.T.)

**Keywords:** African swine fever, virus sequence, epidemiology, genotype II, first case, Sardinia

## Abstract

African swine fever (ASF) is a devastating infectious disease of domestic pigs and wild boar that is spreading quickly around the world and causing huge economic losses. Although the development of effective vaccines is currently being attempted by several labs, the absence of globally recognized licensed vaccines makes disease prevention and early detection even more crucial. ASF has spread across many countries in Europe and about two years ago affected the Italian susceptible population. In Italy, the first case of ASF genotype II in wild boar dates back to January 2022, while the first outbreak in a domestic pig farm was notified in August 2023. Currently, four clusters of infection are still ongoing in northern (Piedmont-Liguria and Lombardy), central (Lazio), and southern Italy (Calabria and Campania). In early September 2023, the first case of ASFV genotype II was detected in a domestic pig farm in Sardinia, historically affected by genotype I and in the final stage of eradication. Genomic characterization of p72, p54, and I73R/I329L genome regions revealed 100% similarity to those obtained from isolates that have been circulating in mainland Italy since January 2022 and also with international strains. The outbreak was detected and confirmed due to the passive surveillance plan on domestic pig farms put in place to provide evidence on genotype I’s absence. Epidemiological investigations suggest 24 August as the most probable time of ASFV genotype II’s arrival in Sardinia, likely due to human activities.

## 1. Introduction

African swine fever (ASF) is a hemorrhagic infectious disease of suids caused by the African swine fever virus (ASFV), a large DNA virus of the *Asfarviridae* family and genus *Asfavirus* [1,2,3]. The ASF genotype I virus was introduced to Europe in 1957, generating the first European wave [4]. The virus was introduced again to Portugal in 1960, spreading to the whole Iberian Peninsula and from this area to both American and European countries. The disease has been successfully eradicated in all these territories except Sardinia, where it has been present since 1978 [4,5,6]. In 2007, ASFV genotype II appeared in Georgia, affecting domestic pigs and wild boar [7]. From 2014, it caused several outbreaks notified in several European Union countries [8]. It currently affects 19 countries on the European continent (Georgia, Armenia, Russia, Azerbaijan, Ukraine, Belarus, Lithuania, Latvia, Estonia, Poland, Moldova, Bulgaria, Hungary, Romania, Slovakia, Serbia, Greece, Germany, and Italy), 11 of them belonging to the EU [9,10,11,12,13,14,15,16,17,18,19,20,21]. Most ASF cases reported in this area are attributed to wild boar, with sporadic outbreaks in domestic pig farms [22]. Eastern European countries are characterized by a different ASF epidemiological scenario, where high numbers of cases are notified in domestic pig populations, mainly in backyard and small farms [23]. In 2018, ASF reached the Asian continent, and in 2019 it reached Oceania and Papua New Guinea. In 2021, the disease reappeared in the Americas after an absence of almost 40 years, having been introduced in the Dominican Republic and later in Haiti [24,25,26].

In January 2022, ASFV genotype II was detected for the first time in mainland Italy [27]. Up to October 2023, genotype II was responsible for 1063 cases in wild boar and 16 outbreaks in domestic pigs (ADNS, as of 2 October 2023). The Italian epidemic spread is currently characterized by long-distance jump transmission from infected regions (i.e., Liguria and Piedmont) to previously ASF-free areas, such as Lazio, Lombardia, Campania, Calabria, and Sardinia [28,29,30]. Transmission via the human factor was confirmed in mainland Italy [29]. Furthermore, after one year of ASF-confirmed wild boar cases, the infection entered intensive pig farms, causing several outbreaks with serious consequences [28,30]. While the epidemic in Italy has worsened, the epidemiological situation in Sardinia in respect of ASFV genotype I appears to be under control and close to eradication [29,31].

From its first detection, ASFV genotype I in Sardinia presented a different epidemiological story than the rest of Europe [31]. The first notification of ASF virus (ASFV) in the Italian island of Sardinia dates to 1978 [5,6]. Since then, the disease has affected three pig populations (domestic pigs, wild boar, and illegal free-ranging pigs) in a typical biological cycle [31]. Of these three populations, illegal free-ranging pigs have been identified as the key population in terms of disease maintenance [32,33,34]. Actions against ASF in the three target populations in Sardinia have been established by the Extraordinary ASF Eradication Program (ASF-EP) on the basis of the epidemiological situation and coordinated by the Project Unit for ASF eradication [31]. Furthermore, the plan has recently included specific actions and controls aimed at avoiding genotype II’s entrance and/or its early detection. It is now possible to confirm that the disease is under control and close to eradication, considering the high level of controls from active and passive surveillance and the absence of ASFV genotype I for more than four years (April 2019) [35]. However, the first genotype II incursion was recently detected in Sardinia, and the first ASFV genotype II outbreak was notified in a small pig farm. Here, we report the description of this outbreak, including the results of epidemiological investigations, diagnosis, and genetic characterization and the main hypothesis in respect of virus introduction.

## 2. Materials and Methods

### 2.1. Characteristics of the Outbreak Farm and ASF Suspicion

The first ASFV genotype II outbreak in Sardinia (an Italian island) was notified on SIMAN on 20 September 2023, in respect of suspected clinical signs and laboratory results from a small farm located in the Dorgali municipality (Nuoro province of Sardinia). As reported in the National Italian Database (BDN), a total of 16 animals were bred in the farm before the outbreak, as reported in Table 1.

Three pigs died on 10 September and were regularly disposed of by the farmer underground with lime as recommended by the ASF official manual [36]. These were considered unsuspicious deaths; thus, no laboratory tests were carried out on the pigs. Nine days later (19 September), three other pigs died, and the farmer alerted veterinary services. On the next day, the veterinary services reported ASF suspicion and carried out stamping out (20 September).

### 2.2. Laboratory Diagnosis

The samples collected on 19 (from three dead pigs) and 20 September (from ten culled pigs), first following outbreak suspicion and subsequently during the culling actions, were sent to the laboratory of the Istituto Zooprofilattico Sperimentale (IZS) della Sardegna for ASFV diagnosis. A total of five spleens, two lymph nodes, two tonsils, four kidneys, two lungs, four sera, and two blood samples were tested for ASFV and ASF antibodies detection. The presence of ASFV antibodies on serum samples was assessed using a commercial ELISA test (Ingezim PPA Compac^®^, Ingenasa, Madrid, Spain) following the manufacturer’s instructions.

Tissues were homogenized 10% in PBS (phosphate-buffered saline). DNA was extracted from clarified supernatant and whole blood samples using Indimag Pathogen IN48 Cartridge (INDICAL) and Indimag 48s extractor, following the manufacturer’s instructions. The presence of ASFV genome was assessed using real-time PCRs approved by the World Organization for Animal Health [36,37,38].

### 2.3. Sanger Sequencing and Phylogenetic Analysis

Molecular typing was performed, analyzing different ASFV genomic regions via Sanger sequencing. The C-terminal region of the B646L gene encoding p72 protein and the complete E183L gene encoding p54 protein were amplified, as previously described [39,40]. The primers ECO1A and ECO1B were used to amplify the I73R/I329L intergenic region [41]. Sanger sequencing was performed on both strands on an ABI-PRISM 3500 Genetic Analyzer (Applied Biosystems, Darmstadt, Germany) with a DNA sequencing kit (dRhodamine Terminator Cycle Sequencing Ready Reaction; Applied Biosystems). The sequences were assembled, edited, aligned to ASFV reference strains retrieved from GenBank, and translated using BioEdit software, version 7.0.0 [42]. The phylogenetic signal of the datasets composed of p72 and p54 sequences was subjected to likelihood mapping analysis with 10,000 random quartets in the TreePuzzle software package, as already described [43]. The evolutionary model that best fitted the data for the datasets analyzed was selected by means of JmodelTest v.2.1.7 [44]. Phylogeny for both p72 and p54 datasets was estimated in MEGA 7 [45] via maximum likelihood and the GTR + G + I model of nucleotide substitution. Statistical support for specific clades was obtained via 1000 bootstrap replicates.

### 2.4. Virus Isolation

Virus isolation was performed according to the hemadsorption (HAD) test described in the OIE Manual [36,46]. Homogenized tissues from all positive samples were added to porcine monocytes/macrophage monolayers, and cells were monitored daily for hemadsorption effects.

### 2.5. Epidemiological Investigation

During outbreak stamping-out, veterinarians carried out an inspection aimed at identifying the origin of ASFV introduction (i.e., epidemiological investigation) [47] and entered the information collected from the ASF outbreak into the Italian National Information System for the Notification of Infectious Animal Disease (SIMAN) database. The epidemiological datasets contain specific information about farm data (i.e., location and farm owner), animals bred (i.e., type of production, number of animals by species), animal census by categories, farm network (i.e., number and type of relationships with other farms), animal movements, external visits to the farm (i.e., veterinarians, breeders, salesmen), clinical evaluation (i.e., number of dead and symptomatic pigs, date of disease suspicion, date of first symptom identification, type of symptoms, number of pigs serologically and virologically tested, number of pigs detected as ASFV-positive and ASFV-antibody-positive). Specific sessions about the epidemiological context included the possible presence of major risk factors, as well as the presence/absence of wild boar near the farm, the natural movement of wild boar, the importation of contaminated pork and pork products, the presence of inadequate biosecurity measures at the farm level and swill feeding, and finally the hypothesis of the most probable origin of the contagion based on the veterinarians’ opinion.

## 3. Results

### 3.1. Laboratory Diagnosis

The serum samples analyzed were negative for ASFV antibody detection. The tissues (three spleens and two kidneys) from the three animals that died on 19 September showed positive results in both real-time PCRs for ASFV genome detection. The Ct (threshold cycle) values ranged between 20.32 and 23.52 using real-time PCR procedure 1 and between 19.6 and 24.87 with real-time PCR procedure 2 described in the OIE Manual [36,37,38]. All the other tissue and blood samples were PCR-negative.

### 3.2. Sanger Sequencing and Phylogenetic Analysis

Molecular typing was performed on the DNA extracted from the three positive spleens, identified as 56187, 56194, and 56196. The p72, p54, and I73R/I329L sequences obtained showed 100% inter-sample identity and were submitted to the GenBank repository as a single record for each genome region (GenBank accession number OR689346-OR689348).

The sequences of the E183L (p54) and B646L (p72) regions had 100% similarity with mainland Italian strains and with reference ASFV genotype II sequences retrieved from GenBank.

The phylogenetic signals of the datasets composed of p72 and p54 sequences evidenced sufficient phylogenetic information, as shown in Appendix A.

The phylogenetic trees depicted in Figure 1A (p72) and Figure 1B (p54) confirm that our sequences clustered with genotype II isolates from different countries, including strains from mainland Italy.

To better differentiate the Sardinian strains, sequencing of the region located between the I73R and I329L genes, characterized by the presence of TRS, was performed. Figure 2 shows the alignment of the I73R/I329L sequences obtained in this study with those of national and international strains belonging to genotype II. Our strains had 100% identity compared with those from mainland Italy [27,48] and other European and Asian countries and differed from Georgia2007 and Armenia07 isolates for one tandem repeat [7,49].

### 3.3. Virus Isolation

ASFV was successfully isolated from all PCR-positive samples. The cytopathic effect was evidenced in spleen samples 15 h post-infection and in the kidney samples within 24 h post-infection.

### 3.4. Case Description and Epidemiological Situation

Given that Sardinia (Italy) is already considered as being infected by ASFV genotype I and is included in Part III of the ASF risk areas (European Commission Implementing Regulation 2021/605/EU), the infected farm falls in zone I of the ASFV genotype I restrictions. As reported by the “European Commission Implementing Decision concerning certain interim emergency measures relating to African swine fever in Italy”, protection and surveillance areas with radii of 10 and 15 km, respectively, were established around the outbreak for three months (until 22 December), as illustrated in Figure 3. This decision was based on the Terrestrial Animal Health Code (OIE, Cap. 15.1, Articles 15.1.6 and 15.1.7), which establishes the recovery of the status of a previously ASF-free country.

As shown in Table 2, upon entering the farm, the veterinarians noted the deaths of the first three animals (10 September) and observed other three dead pigs (19 September). Then, they identified 10 live animals: one sow, one fattening pig, and eight piglets. All these live animals were culled and regularly disposed of on 20 September by the veterinary authorities during the inspection.

The clinical history reported by the veterinarians and the farmer described typical ASFV lesions [50] as well as unexpected death, hemorrhaging of the digestive tract, renal petechiae, and enlarged spleens filled with blood. The organs and blood sera from the three animals that died on 19 September and the other ten pigs culled on 20 September were sent to the IZS laboratory of Sardinia for ASFV diagnosis.

As reported after the epidemiological investigation, no new animals had been introduced to or sold from this farm since the last check for ASF (31 March 2023, ASF serological tests), as planned by the ASF Eradication Plan. Only the farmer had taken care of the animals, and no external people had visited the farm or met the animals. The farm is characterized by high biosecurity measures: barriers and fences that avoid contact with wild animals or unauthorized people and all pigs kept in closed areas. All animals were correctly identified, and the farm’s registry was correctly compiled. However, the farmer did not quarantine newly introduced animals, clothing was not of single use, and the use of disinfectants against ASFV was not evident. Furthermore, the farmer declared that they have a butcher shop and work in a slaughterhouse, and the animals are fed with food waste.

### 3.5. Hypothesized Origin of the Infection

Considering that the infected farm was categorized as a certified farm [31] with high biosecurity, never correlated with previous outbreaks, the most probable method of ASFV introduction is attributable to the administration of food waste.

To support this hypothesis, and based on the main risk factors (i.e., the farmer is the owner of a butcher shop and works in slaughterhouses), the local veterinarians traced the origin of the meat with which the farmer had come into contact. The tracing took into account the incoming movements of live pigs and pig products between 20 July and 18 September 2023. The tracing is illustrated in Figure 4.

To ensure anonymity, the wholesalers, stockists, or retailers involved in the infected meat trade are named “actors”. The geographical focus of the infected meat and meat products trade is illustrated in Figure 5.

As reported by several Italian newspapers, the farmer and the local veterinarian of a pig farm located in Zinasco (Pavia province, Lombardia region, mainland Italy) would not have reported the first suspicious deaths on the pig farm [51]. Based on this hypothesis, the Pavia Prosecutor’s Office opened an investigation into the outbreak [52]. Until 29 August, when the Regional Authorities discovered the outbreak and started culling, animals continued to be sent to the slaughterhouse from this farm. The first hypothetical jump of the virus is represented by the trade from Actor A (infected pig farm in Zinasco) to Actor B (wholesalers of Cesena (Forlì-Cesena province, Emilia-Romagna region, mainland Italy)).

Subsequently, the main hypothesis is the long-distance jump of the virus from B to C1, C2, and C3: on 23 August, the meat products were sold from the wholesalers of Cesena to three Sardinian retailers, located in the northwest, northeast, and south of the island. The meat products arrived on the island on 24 August and were claimed back on 29 September by Actor B. Actors C1 and C3 claimed back all the products (118.22 kg and 12.94 kg, respectively), while C2 claimed back only 80.26 kg. The remaining 219.80 kg of infected meat was sold all over the Sardinian region, as illustrated in Appendix A Appendix A. This 219.80 kg of meat was notified as consumed without residuals by local retailers. One of these sales probably caused the outbreak of ASFV genotype II in Dorgali (Nuoro province of Sardinia). After the purchase of the meat products by C2, Actor D2 fed his pigs with food waste (probably including the infected meat) on 26 August. After exactly 15 days (10 September), the first three pigs died and subsequently another three died on 19 September. On the same day, the farmer called local veterinarians, who undertook culling, and the outbreak was notified. Between 19 and 20 September, samples from culled animals were collected and tested, and the results were disclosed to the local authorities. The culled pigs tested virologically and serologically negative. Thus, a recent intra-farm infection that was immediately stopped can be hypothesized.

On 7 September, the Cesena Local Sanitary Agency, where Actor B is located, communicated the risk of ASFV-infected meat or meat product introduction to the Local Sanitary Agency of Sardinia. The official report communicated by the Regional Sanitary Agency of Emilia Romagna to the Regional Sanitary Agency of Sardinia dates back to 21 September, when the outbreak had already been managed and closed. Otherwise, this method of contagion is only hypothesized, given that the meat supposedly infected was not sampled and tested for ASFV.

## 4. Discussion

Sardinia had been affected by ASFV genotype I for more than 40 years [53]. During this period, only one case of the virus escaping from the island was notified in 1983, through the movement of meat products [31]. Less than two years from the first genotype II notification in Italy, the virus crossed the sea and infected a domestic pig farm in Sardinia.

Thanks to the ASF Eradication Plan that is in place and the stringent measures to contrast genotype I, the Sardinian veterinarian control system immediately detected and managed the outbreak. Particularly in 2022, when Sardinia’s status was modified from Part IV to Part III of the ASF risk areas (European Commission Implementing Regulation 2021/605/EU) [35], a specific plan for passive surveillance in domestic pig farms was implemented. The plan consists of sampling a representative number of dead pigs in each sanitary local area, and the sampling of dead pigs with suspected death from ASF is notified by the farmer (i.e., high mortality in the farm). The immediate identification of the ASFV genotype II outbreak in dead pigs confirms the effectiveness of the plan in respect of early detection of the virus.

Additional staff have been provided to support the veterinary services responsible for these control actions. For biosecurity reasons, veterinary teams and technicians who came into contact with the outbreak were prevented from participating in control actions for the following weeks. The correct management of the outbreak will also be determined by the speed of controls within the two restriction areas, an activity that will allow us to avoid the danger of possible secondary outbreaks and therefore the spread of the virus within larger territories [23]. The interventions and measures referred to in the restriction zones were issued by the competent authority (Local Sanitary Agency of Nuoro) on 20 September. These activities were carried out within the subsequent 15 days. The actions put in place within the restriction zones consisted of clinical visits and laboratory tests for the early detection of the virus. During the execution of the clinical visits, any clinical symptoms of disease were highlighted, blood samples were taken to verify the presence of the virus through direct diagnostic methods, and any non-compliance with biosecurity and traceability of the animals were recorded.

Indeed, considering the absence of ASFV since 2019 [35,54], an alert campaign for veterinarians and farmers was put in place, particularly to raise awareness about the notification of suspected deaths of pigs inside farms. If the deaths exceed 20% for one breed of pig, clinical inspection and specific serological and virological tests must be performed to exclude the presence of ASFV. Even though passive surveillance of wild boar has been in place since 2020 [35], considering that the farmer of the outbreak farm is also a hunter, specific passive surveillance activities were reinforced. People and dogs specialized in wild boar carcass searching scoured the land around the farm and in the field visited by the farmer and his dogs between 25 August and 19 September.

Furthermore, early disease detection was possible thanks to the cooperation of the farmer. It has been repeatedly demonstrated how social factors, raising awareness among farmers, and collaborating with them are key factors in the fight against ASF [6,55,56]. The farmer demonstrated great trust in the veterinarians and good collaboration with them; he notified the veterinary services and demonstrated that the regional authorities have built a relationship of trust with pig farmers over the years, allowing us to fight the disease adequately.

The genomic characterization performed by sequencing the p72, p54, and I73R/I329L genome regions allowed us to assign the strains under study to ASFV genotype II. All sequences had 100% similarity with those obtained from isolates that have been circulating in mainland Italy since January 2022 but also with international strains [48,49]. Therefore, this first genotyping approach does not help trace the exact geographical origin of the virus, supporting the results of the epidemiological investigation.

At this stage, the epidemiological situation in several areas of Italy is complicated by the difficulties in contact tracing. As often happens when a highly contagious disease epidemic starts, its rapid spread hampers the fast tracing of each contact [57], favoring virus diffusion. Furthermore, the long-distance jump over the sea by ASFV genotype II cannot be underestimated. Considering the enormous effort put in place by Sardinia to eradicate ASFV genotype I, the current priority should be to protect the island from genotype II. Further evaluations should be carried out to identify the most probable method of introduction of the virus and allocate resources to areas at higher risk, improving prevention and control strategies and promptly stopping ASFV. Even though Mugnoz-Perez [58] described the risk of ASF introduction in Spain via the legal importation of swine products as very low (1.74 × 10^−4^), this is probably not valid for Italy or any country involved in the early phases of disease introduction.

## 5. Conclusions

Currently, the future development of secondary cases or further virus incursion in Sardinia cannot be excluded, and the strong control measures put in place must be retained until the end of the protection and surveillance zone period (22 December 2023). The successful and prompt management of the outbreak indicates the high experience level of Sardinia to face ASF and demonstrates that expertise makes a difference in disease control. Full-genome sequencing could help in finding genetic markers capable of discriminating among closely related viruses and support epidemiological investigation in tracing the movement of the virus.

## Figures and Tables

**Figure 1 viruses-16-00032-f001:**
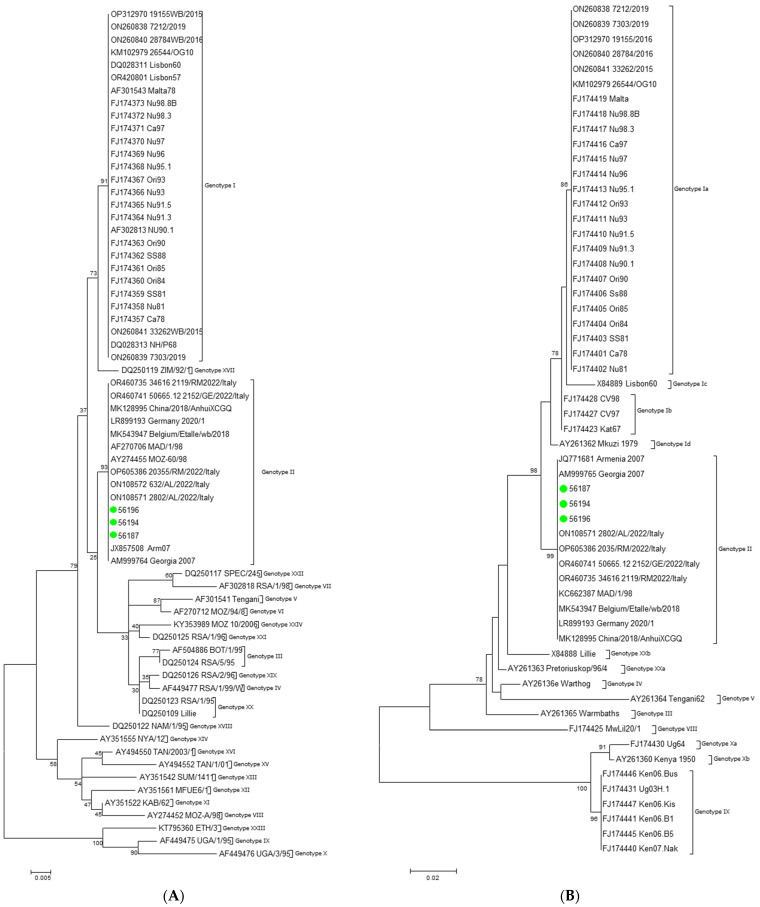
A maximum likelihood phylogenetic tree inferred from the dataset composed of p72 (**A**) and p54 (**B**) sequences by the GTR+G+I model of nucleotide substitution. Strains under study are indicated with green circles. Bootstrap values < 70 are not shown. The scale bar indicates the number of substitutions per site.

**Figure 2 viruses-16-00032-f002:**
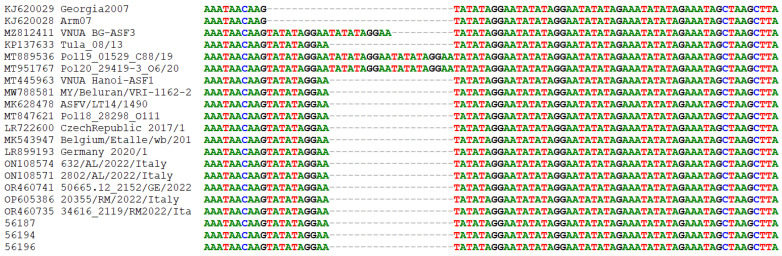
Alignment of I73R/I329L intergenic regions from Italian and international ASFV genotype II strains.

**Figure 3 viruses-16-00032-f003:**
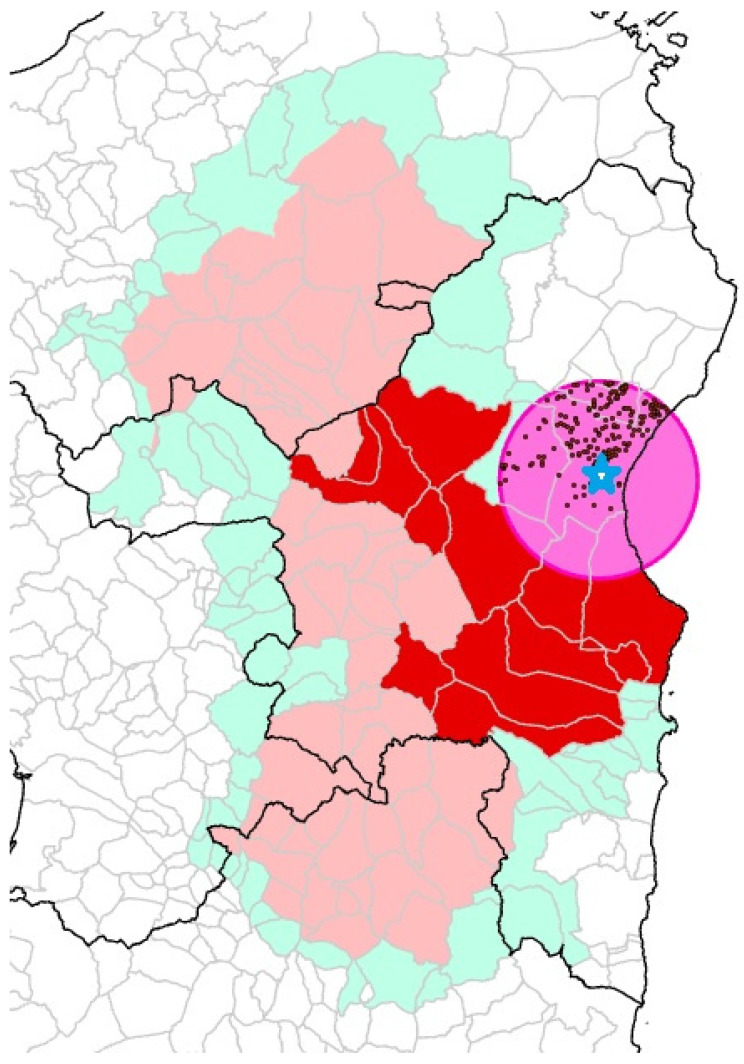
Location of the outbreak farm (blue star) and the pig farms (black dots) located in the protection and surveillance area (pink) around the outbreak with respect to restriction zones I (light blue), II (light pink), and III (red) for ASFV genotype I in Sardinia.

**Figure 4 viruses-16-00032-f004:**
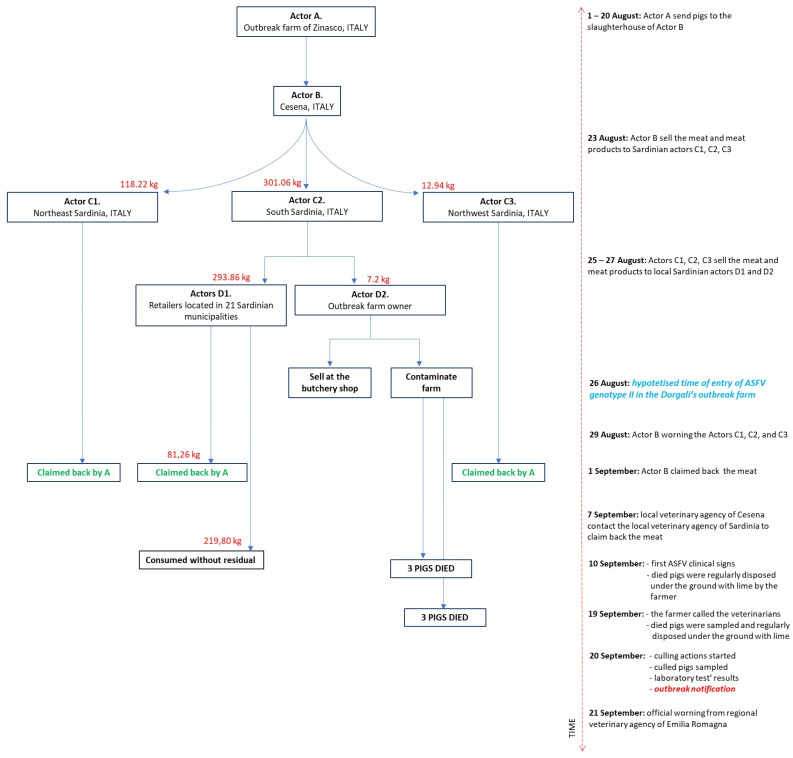
The figure illustrates the tracing of the meat and meat products that hypothetically caused the ASFV genotype II outbreak. On the left, the flow chart indicates each actor involved; on the right, the time and actions are described.

**Figure 5 viruses-16-00032-f005:**
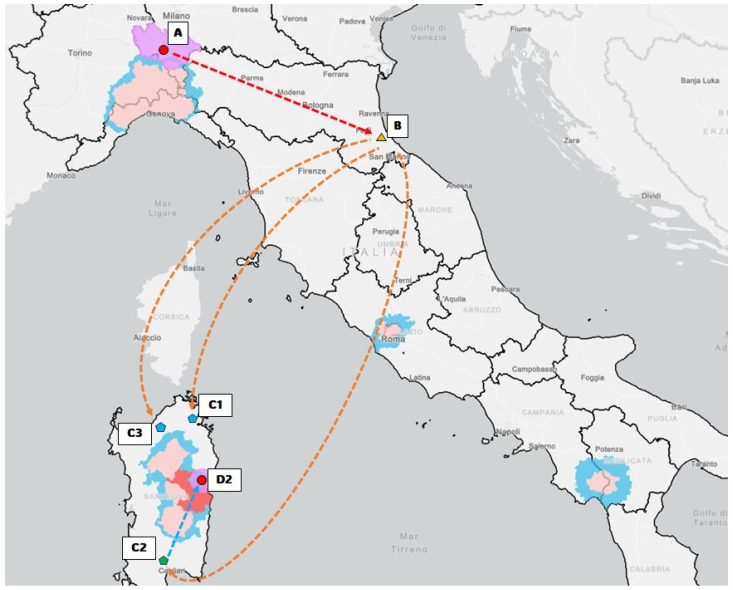
This figure geographically illustrates the tracing (arrows) of the meat and meat products that hypothetically caused the ASFV genotype II outbreak. Actors A (red dot), B (yellow triangle), C1 (blue pentagon), C2 (green pentagon), C3 (blue pentagon), and D2 (red dot) refer to farms in Zinasco (Pavia province, Lombardia region), wholesalers in Cesena (Forlì-Cesena province, Emilia-Romagna region), retailers in Northwest Sardinia, retailers in Northeast Sardinia, retailers in South Sardinia, and a farmer in Dorgali, respectively. The blue areas indicate level I restriction zones, the pink areas indicate level II restriction zones, the red area indicates the level III restriction zone, and the violet areas indicate the protection and surveillance areas around outbreaks.

**Table 1 viruses-16-00032-t001:** Census by category and distribution of pigs bred on the farm.

Pig Categories	Total Pigs before the Outbreak
Male for reproduction	1
Sow	4
Fattening pig	3
Piglet	8
Total	16

**Table 2 viruses-16-00032-t002:** Census by category and distribution of pigs bred on the farm and culled during the stamping out.

Pig Categories	Dead Pigs	Culled Pigs ^1^	Total Pigs
Male for reproduction	1	0	1
Sow	3	1	4
Fattening pig	2	1	3
Piglet	0	8	8
Total	6	10	16

^1^ Live pigs that were culled during the inspection.

## Data Availability

All the data are reported in the main text.

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
