# Peer review of "The Long-Jumping of African Swine Fever: First Genotype II Notified in Sardinia, Italy"

_viruses, 2023, doi:10.3390/v16010032_

Round 1

Reviewer 1 Report

Comments and Suggestions for Authors

This manuscript describes first report of ASFV p72 genotype in Sardinia. The manuscript is timely and is of important to ASF researchers and policy makers.

I recommend accepting the manuscript for publication after major revisions.

1. Extensive editing in English is required. Incorrect grammar, informal language, unclear statements throughout the manuscript.

2. Some information is repeated multiple times throughput the   manuscript. Please revise to avoid repetition and keep the flow

2. Under Materials and Methods : Farm information needs a sub heading. Not clear from which animals serum was collected. It is better to describe the composition of the farm at the sampling here.

3. Under epidemiological investigation, when discuss the locations it is better to mention if it is in main land or in Sardinia, because many readers are not familiar with the cities in Italy and Sardinia.

4. The section 3.4 (Control measures put in place in Sardinia subsequently the first genotype II outbreak) can be moved to the discussion.

5. Authors mention " The 29th August the Regional Veterinary Authorities of Pavia discovered the infection 294 in the farm of Zinasco (Pavia, Lombardia), where the culling action were not put in place 295 and infected animals not culled [52,53]" This needs to be discussed more, why no culling was done in this farm. This could be a translation issue.

6. Epidemiological data strongly suggest that  genotype II  virus spread from mainland Italy to Sardinia in contaminated pork products but the sequence data presented here is not adequate to confirm that.  Therefore, authors  must take precaution making strong statement. Whole genome sequencing is required to make a direct link.

7. Too many figures in the main body of the manuscript. I suggest combining and moving some figures to Appendix

Figure 2  can be a Supplementary figure 

Figure 4 and 6 cane be combined. Figure 4 can be an inlet in Figure 6

Figure 7 can be a Supplementary figure 

Comments on the Quality of English Language

Extensive editing in English is required. Incorrect grammar, informal language, unclear statements through out the manuscript  

Author Response

Response to Rev.1

This manuscript describes first report of ASFV p72 genotype in Sardinia. The manuscript is timely and is of important to ASF researchers and policy makers. I recommend accepting the manuscript for publication after major revisions.

  1. Extensive editing in English is required. Incorrect grammar, informal language, unclear statements throughout the manuscript.

R: the manuscript has been completely reviewed by the MDPI English editing service.

  1. Some information is repeated multiple times throughput the manuscript. Please revise to avoid repetition and keep the flow

R: the repetitions were deleted through the manuscript

  1. Under Materials and Methods: Farm information needs a sub heading. Not clear from which animals serum was collected. It is better to describe the composition of the farm at the sampling here.

R: a new sub heading has been created named “Characteristics of the outbreak farm”

  1. Under epidemiological investigation, when discuss the locations it is better to mention if it is in main land or in Sardinia, because many readers are not familiar with the cities in Italy and Sardinia.

R: specification about Italy mainland or Sardinia were already included at the first citation of each location, Anyway, considering the opinion of the revisor, this has been included at each time in the text.

  1. The section 3.4 (Control measures put in place in Sardinia subsequently the first genotype II outbreak) can be moved to the discussion.

R: the section was moved

  1. Authors mention " The 29th August the Regional Veterinary Authorities of Pavia discovered the infection 294 in the farm of Zinasco (Pavia, Lombardia), where the culling action were not put in place 295 and infected animals not culled [52,53]" This needs to be discussed more, why no culling was done in this farm. This could be a translation issue.

R: It is not a translational issue. The sentence has been deleted because a repetition of lines 265-268

  1. Epidemiological data strongly suggest that  genotype II  virus spread from mainland Italy to Sardinia in contaminated pork products but the sequence data presented here is not adequate to confirm that.  Therefore, authors  must take precaution making strong statement. Whole genome sequencing is required to make a direct link.

R: we completely agree with this point of view, in fact in discussion the authors affirm: “The genomic characterization performed by sequencing the p72, p54 and I73R/I329L genome regions allow us to assign the strains under study to ASFV Genotype II. All sequences had 100% of similarity with those obtained from the isolates that have been circulating in mainland Italy since January 2022, but also with international strains [48–50]. Therefore, this first genotyping approach doesn’t help tracing the exact geographical origin of the virus supporting the results of the epidemiological investigation.“ and in conclusions affirm “The full genome sequencing could help finding genetic markers capable of discriminating among closely related viruses and support the epidemiological investigation in tracing the movement of the virus.”

  1. Too many figures in the main body of the manuscript. I suggest combining and moving some figures to Appendix

Figure 2 can be a Supplementary figure

R: done

Figure 4 and 6 cane be combined. Figure 4 can be an inlet in Figure 6

R: dear revisor, even if we completely understand your point of view, we do not agree to combine these two figures because we believe that the result would be too complicate and difficult to understand by the readers. Please, let us know if this is a fundamental point of your revision, or if we can find a solution together.

Figure 7 can be a Supplementary figure 

R: moved

Reviewer 2 Report

Comments and Suggestions for Authors

This manuscript presents an interesting evaluation of the recent introduction of genotype II ASFV to the island of Sardinia. Sequencing of genome fragments including some more variable regions indicates mainland Italy as the likely source and epidemiological tracing defined the events leading to introduction of infected pork and spread of ASFV to pigs by feeding infected pork. The results are of interest to those interested in ASFV epidemiology and mechanisms of long distance spread. The research was carried out thoroughly enabling early detection and hopefully eradication of this strain of virus from Sardina. 

Comments on the Quality of English Language

The language needs extensive editing

Author Response

the manuscript has been completely reviewed by the MDPI English editing service

Reviewer 3 Report

Comments and Suggestions for Authors

The article "Long jump of African swine fever in Italy: first genotype II notified in Sardinia" by Silvia Dei Giudici et al., is the first report of genotype II African swine fever virus in Sardinia. The main objective of the article was to report the first case of genotype II ASFV in Sardinia, starting from its first detection and genomic confirmation in the laboratory to tracing back to the origin in Italy and present the most probable hypothesized route of the long jump from Italy to Sardinia using the critical epidemiological investigations conducted by the Sardinian veterinary authorities. 

Apart from multiple grammatical and spelling corrections that need to be addressed extensively in the text, the manuscript was well written and certainly delivers the story clearly. It's valuable for the ongoing efforts in the control of ASF in the world.

While the genomic characterization was based on partial gene sequencing of P72, P54 and I73R/I329L genes at this time, and it was enough to determine the presence of genotype II and hypothesize the potential source of infection, the authors understand the importance of whole genome sequencing for further characterizing the virus to support the epidemiological investigation.

The prompt control measures and the critical investigations conducted by the Sardinian veterinary authorities to control and prevent further spread of the virus is commendable.

Comments on the Quality of English Language

There are multiple spelling and grammatical corrections that need to be done. I'm requesting minor revisions to correct the English. 

Author Response

Dear review, thank you very much for your support and suggestions. As you suggested, the manuscript has been completely reviewed by the MDPI English editing service

Round 2

Reviewer 1 Report

Comments and Suggestions for Authors

The English language of the manuscript have improved significantly, however I kindly  request  the authors to consider making following changes to the title and abstract.

Title: . long-jumping implies spread of ASF from main land to Sardinia. This is highly possible but no molecular evidence at this time. Therefore I suggest a simple title like the one below

Detection of African swine fever  p72 genotype II in Sardinia

For example, in the abstract 

1. The first case in wild boar dates to January 2022, while the first outbreak in a domestic pig farm was notified in August 2023.

Should be

In Italy, the first case of ASF genotype II in wild boar dates back to January 2022, while the first outbreak in a domestic pig farm was notified in August 2023.

2. Currently, four clusters of infection are still ongoing in the north (Piedmont-Liguria and Lombardy), center (Lazio), and south of Italy (Calabria 25 and Campania).

Should be

Currently, four clusters of infection are still ongoing in the northern (Piedmont-Liguria and Lombardy), central (Lazio), and southern Italy (Calabria  and Campania).

 3. In early September 2023, the first overseas long-jump of the virus was detected in a domestic pig farm in Sardinia, historically affected by genotype I and in the final stage of eradication.

This implies virus jumped from Italy to Sardinia. Although epidemiological information support this argument, genomic data is lacking to support this.

4. Added the first case of ASFV genotype II   to clarify the sentence - In early September 2023, the first case of ASFV genotype II  virus was detected in a domestic pig farm in Sardinia, historically affected by genotype I and in the final stage of eradication.

Revise the rest of the abstract as follows to maintain the flow and clarity.

Genomic characterization of  p72, p54, and I73R/I329L genome regions revealed  100% similarity to those obtained from isolates that have been circulating in mainland Italy since January 2022, and also with international strains. The outbreak was detected and confirmed due to the passive surveillance plan on domestic pig farms put in place to provide evidence on genotype I absence. Epidemiological investigations suggest the 24th August as the most probable time of ASFV genotype II  arrival in Sardinia, likely due to human activities.

Comments on the Quality of English Language

The English language of the manuscript have improved significantly, however I kindly  request  the authors to revise the abstract as requested

Author Response

The English language of the manuscript have improved significantly, however I kindly  request  the authors to consider making following changes to the title and abstract.

R: dear reviewer, thank you very much for the time spent to review next time our manuscript. Your kind requests improved the paper and help the reader in understanding the work. Otherwise, we are so afraid to the problems with the English language that you highlight and we’ll certainly make our grievances to the MDPI editing service for the money and the time spent.

Title: . long-jumping implies spread of ASF from main land to Sardinia. This is highly possible but no molecular evidence at this time. Therefore I suggest a simple title like the one below

Detection of African swine fever  p72 genotype II in Sardinia

R: dear reviewer, we completely understand this point of view, as several time reported across the manuscript, particularly in the discussion. We change the title with:

The long-jumping of African swine fever: first genotype II notified in Sardinia, Italy.

The title we proposed is more attractive but maintaining the mean you suggested.

For example, in the abstract 

  1. The first case in wild boar dates to January 2022, while the first outbreak in a domestic pig farm was notified in August 2023.

Should be

In Italy, the first case of ASF genotype II in wild boar dates back to January 2022, while the first outbreak in a domestic pig farm was notified in August 2023.

 R: corrected as you suggested

  1. Currently, four clusters of infection are still ongoing in the north (Piedmont-Liguria and Lombardy), center (Lazio), and south of Italy (Calabria 25 and Campania).

Should be

Currently, four clusters of infection are still ongoing in the northern (Piedmont-Liguria and Lombardy), central (Lazio), and southern Italy (Calabria and Campania).

 R: corrected as you suggested

  1. In early September 2023, the first overseas long-jump of the virus was detected in a domestic pig farm in Sardinia, historically affected by genotype I and in the final stage of eradication.

This implies virus jumped from Italy to Sardinia. Although epidemiological information support this argument, genomic data is lacking to support this.

 R: corrected as you suggested

  1. Added the first case of ASFV genotype II to clarify the sentence - In early September 2023, the first case of ASFV genotype II virus was detected in a domestic pig farm in Sardinia, historically affected by genotype I and in the final stage of eradication.

Revise the rest of the abstract as follows to maintain the flow and clarity.

Genomic characterization of p72, p54, and I73R/I329L genome regions revealed 100% similarity to those obtained from isolates that have been circulating in mainland Italy since January 2022, and also with international strains. The outbreak was detected and confirmed due to the passive surveillance plan on domestic pig farms put in place to provide evidence on genotype I absence. Epidemiological investigations suggest the 24th August as the most probable time of ASFV genotype II arrival in Sardinia, likely due to human activities.

 R: corrected as you suggested